# Research on the Impact of Tourism Development on the Sustainable Development of Reservoir Headwater Area Using China's Tingxi Reservoir as an Example

**Chih-Chien Shen** [1] [ID]**, Chou-Fu Liang** [2,*]**, Chin-Hsien Hsu** [3]**, Jung-Hul Chien** [4] **and Hsiao-Hsien Lin** [3,*] [ID]

[1]  Institute of Physical Education and Health, Yulin Normal University, Yulin 537000, China; g169168@gmail.com
[2]  School of Environmental and Life Sciences, Nanning Normal University, Nanning 530001, China
[3]  Department of Leisure Industry Management, National Chin-Yi University of Technology, Taichung 41170, Taiwan; hsu6292000@yahoo.com.tw
[4]  Department of Social Work, Toko University, Chiayi 61249, Taiwan; chien0106@yahoo.com.tw
*  Correspondence: Lcf66131@yahoo.com.tw (C.-F.L.); chrishome12001@yahoo.com.tw (H.-H.L.)

**Abstract:** The purpose of this study was to understand the impact of tourism development on the sustainable development of Tingxi Reservoir. Based on tourism impact theory, 804 questionnaires were statistically validated and analyzed, followed by a semi-structured interview with five respondents, and finally examined by a multivariate verification method. The study found that not only did development fail to raise land and housing prices, develop leisure activities, improve medical facilities, and supplement police manpower, but it also increased consumer costs and environmental damage. There were also problems such as insufficient interpreters, parking and rest facilities, and ineffective management of communication channels, bicycle facilities, and tourist waste, which did not help youths to return to their hometowns. Furthermore, due to the disparities in the performance of leisure opportunities, medical and health care, spatial planning, and cultural development, there were different opinions among the stakeholders. Suggestions: (1) Satisfy the needs of different stakeholders; (2) Improve the environmental literacy of tourists and provide more garbage cans; (3) Develop additional scenic spots to divert tourists; (4) Stabilize prices and attract investment from enterprises; and (5) Increase the participation of residents in community development to supplement industrial manpower.

**Keywords:** multifunctional water source area; ecotourism, people with different stakeholders; balanced decision-making

---

## 1. Introduction

Tingxi Reservoir was built in 1956 as an artificial lake in Tong'an District, Xiamen City, Fujian Province, China. In addition to providing water storage and irrigation, the surrounding area features diverse natural ecology, rich catches, numerous cultural sites, Song dynasty porcelain, and historical relics [1], so the nearby villages are striving to build and develop tourism resources in order to improve development and realize the goal of revitalizing the village economy. The area has now become a lake and water resource area with a variety of functions including water storage, drinking water, flood control, power generation, irrigation, aquaculture, art, culture, archaeology, fishery, livestock, hot springs, and forestry resources [2], as shown in Figure 1. It is estimated that it attracts 777,800 visitors each year, and it has set a record of 1,013,300 visitors in seven days, generating US$34,146,000 in revenue [3], which shows the effectiveness of tourism development.

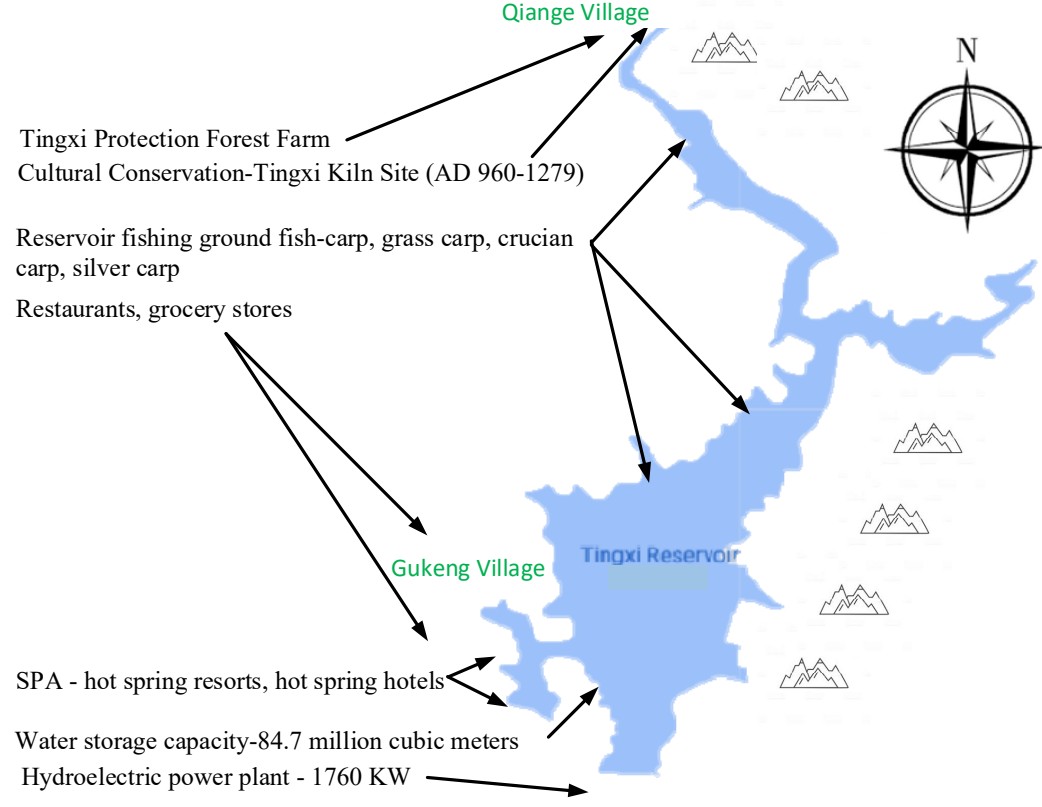

**Figure 1.** Distribution map of the tourism resources at Tingxi Reservoir.

However, policy decisions affect the direction and effectiveness of tourism development [4,5]. While local development can bring positive improvements in the living standards of the people and environmental sanitation, it can also have negative effects [6,7], destroying existing environments or cultures. In addition, tourism development can bring about changes. Overall, the impact of development can be explored from three levels: economic, social, and environmental [8,9].

The construction of the reservoir source area is not only to provide livelihood and industrial development, but also to bring economic improvement to the local village after proper planning. However, changes in the economic level may have positive and negative effects. Basically, we can get a comprehensive answer from the perspective of the cost of living, business development, economic infrastructure, village development, etc. [10,11]. However, researchers believe that economic development can promote business interaction, increase local tax revenue, and provide funds for improving local public facilities. Therefore, we focused on the cost of living, industrial construction, and community development as the main direction of investigation, and then carried out an in-depth exploration of the development status of local medical care, employment, wages, consumption, construction, industry, facilities, prices, concessions, sanitation, cultural and creative industries, rewards, leisure activities, community feedback, and policy coordination [10–13], so that we can understand the actual extent of changes caused by development to the economy.

The construction of a reservoir is tantamount to altering the appearance of an existing ecological and human habitat on a large scale. Although moderate destruction can create better living conditions and bring about social stability, the rights of the original villagers to survive will be sacrificed. According to previous studies, sustainable development should take into account the needs and expectations of the local villages and people. Therefore, the impact of the reservoir development on the society is relatively diversified, and we should be able to understand the needs of the local people by looking at tourism facilities, community development, living atmosphere, culture and customs, fire prevention and security [10,14], etc. However, the unique culture and customs of local villages provide an image of stability to the people, so that they can live in a pure and simple manner, stabilizing their living

conditions and reducing the occurrence of security incidents. Furthermore, local development is based on the premise of improving the local community and people's lives to create a better living environment. Therefore, with the village development, living atmosphere as well as culture and security as the main directions of investigation, we then looked into issues such as local reputation, quality of services and activities, policy participation, tourism organization planning, tourism indicators, cultural and architectural characteristics, security maintenance, community construction, interaction among people, and youth returning to their hometowns [14–17], through which we believed that we could see how development has changed the villages.

Reservoir development changes the existing natural environment and ecology to achieve the original purpose of reservoir construction and improve people's livelihood and economy, but at the same time, it affects the entire society and creates a conflict between local social benefits and losses, but more importantly, it changes the original natural and ecological environment. Therefore, from the analysis of tourism facilities and natural ecology [10,11,14], it should be possible to get a full picture of the impact of development on the environment. However, development not only causes changes in the appearance of the water source area, but also in the current state of local human and ecological development. Furthermore, construction is a type of change and should be a change for the better, to better meet the needs of human beings and to make the existing natural landscape and ecology more sustainable. Therefore, from the perspective of conservation measures, leisure environment, tourism facilities, landscape and ecological environment, we can look into issues such as public transportation, parking and open space, Internet communication, monuments and buildings, residents' environmental awareness, visitors' environmental quality, garbage volume, forest and ecological habitat, motor vehicle fumes, water source, and air quality [14–21], which is helpful to understand the impact of development on the existing environment.

The current development of water sources is moving toward a multifunctional development and management concept based on sustainable development. However, change can be tangible or intangible, and with the accumulation of time [12,13], the most realistic feelings of long-time residents can be recorded with their eyes and body perceptions [10–15]. As a matter of fact, however, it is not objective to describe the merits of development decisions from a single perspective, although the residents' perceptions may present the most realistic picture. This is because development not only aims at preserving the original functions and maintaining the existing appearance, but also aims at making the original facilities or resources more valuable and satisfying more human needs [11,22]. After all, the number of natural, socio-cultural, and economic resources needed to be obtained, and the level of consideration are different for different positions and roles [14,23]. For existing residents, development is expected to improve their quality of life by improving economic conditions, sanitation, transportation, and medical facilities, but they do not want to destroy their original living style and environment [24]. For tourists, water source areas are one of the most attractive tourism resources due to their ecological diversity and the differences in the customs and cultures of existing villages [11], and fulfilling their psychological needs through tourism [12,25] will satisfy tourists and strengthen their tourism and consumption behaviors. It can be seen that both the residents and tourists expect the development of water resource areas to be effective, but they are afraid of the gap between the results and their expectations.

In order to strike a balance, the current tourism research and development should be considered from the perspective of both residents and tourists. As development is aimed at improving local conditions, and tourism development is aimed at attracting tourists and boosting the local economy [6], it takes time to prove the effectiveness of development, and policymakers need the assistance of residents in order to achieve success. Therefore, the most effective way to understand the development process and its effectiveness is to conduct a review with the residents as the subject [6,26]. However, development is the betterment of the existing predicament, while tourism development hopes to attract tourists to travel and spend, in order to achieve the purpose of boosting the economy, increasing people's income, raising revenue, and improving the quality of life of the villages and people [24]. Therefore,

the more time tourists spend on in-depth tourism, the more the desire and opportunities for consumption can be stimulated. It shows that the effectiveness of tourism development should be based on the needs and feelings of residents and tourists [11,14].

Taken together, a certain degree of tourism development in Tingxi Reservoir can satisfy the needs of both residents and visitors, but it can also lead to rejection. By conducting a survey based on environmental issues such as the community environment, the villagers' environmental literacy, preservation of historical sites, awareness of ecological conservation, coordination of conservation policies, maintenance of tourist trails and bicycle lanes, tourism transportation planning, Wi-Fi network speed, bicycle rental, community modernization and scale, rest and parking space, experience of tourism activities, water and air quality, fumes from steam and locomotives, mountain slope development, etc. can be determined [14–20]. Social issues can be identified such as tourism visibility, quality of services and activities, content of community activities, friendly treatment of village culture, tourism indicators or descriptions, tourism and leisure facilities, human resources, DIY activities, hardware and software facilities, tourism environment and space quality, indigenous cultural traditions and historical relics, the image of tourism companies or organizations, the promotion of traditional cultural activities, the interaction mechanism between villagers and tourists, the number and popularity of traditional culture and characteristic industries, the management and safety maintenance of activities, the allocation of service or management personnel, the sense of travel security, the willingness to travel again, etc. [10,14–17]. Economic issues such as employment and entrepreneurship opportunities, tourism activity prices, tourism consumption costs, increased tourism facilities and local characteristic industrial products, tourism diversification, provision of explanatory guides, increase leisure and life consumption options, obtain promotion or priority use rights, and expand community tourism. Through the scale of development, the quality of public facilities construction and maintenance, the quality of medical and health services, the communication channels between the community and the government, the protection policies of the local tourism industry, the mechanisms and norms involved in formulating tourism development policies, the development of DIY or product portfolios, etc., it should be possible to obtain the feelings of both sides and to explore the most in-depth issues to obtain a balanced view for future decision-making [10,11,14,24].

Until now, however, the majority of reservoir and lake studies have been conducted from the perspective of residents [27–29], while fewer have been conducted from the perspective of tourists [30–32], and even fewer have been conducted from the perspective of both residents and tourists [33,34]. Most of the studies that focus on Tingxi Reservoir are about water quality and ecology, water flow, water quality, and power generation [35–37], and almost no scholars have focused on tourism issues. Therefore, the researchers believed that by examining the current tourism development of Tingxi Reservoir from the perspective of residents and tourists, the latest and most realistic answers could be obtained for the reference of local organizations and people. The purpose of this study was to: (1) understand residents' awareness of the current tourism development of Tingxi Reservoir; (2) explore tourists' awareness of the current tourism development of Tingxi Reservoir; and (3) analyze the difference in awareness between residents and tourists on the tourism development of Tingxi Reservoir.

## 2. Methods and Instruments

### 2.1. Study Framework and Hypotheses

The literature finds that most studies on reservoirs and lakes are conducted from the perspective of residents [27–29], while the research conducted from the perspective of tourists [30–32] is rarely conducted, from the perspective of residents and tourists. There is even less discussion [33,34]. Most of the research on Tingxi Reservoir involves water quality and ecology, water flow, water quality and power generation [35–37], and almost no scholars pay attention to tourism issues. Therefore, the researchers believe that the latest development results and dilemmas can be obtained by examining the tourism development status of Tingxi Reservoir from the perspective of residents and tourists.

We sought to collect the experiences of two different groups including residents and tourists on the tourism development of Tingxi Reservoir, point out the current problems, and make suggestions for improvement. The research methods and tools were determined according to the existing information of the case and the relevant literature on the reservoir [1,2,35,36] and the lake and other tourism development issues [3–34]. By applying tourism impact theory and combining the opinions of residents and visitors [10,11,14,16,21,38], we used methods of surveys, interviews, and observations to collect research information [39], and then by comparing and verifying the data [40], we used induction, organization, and analysis sequences to construct this paper [16], in order to obtain correct and reasonable information to revise the development plan of Tingxi Reservoir. As shown in Figure 2.

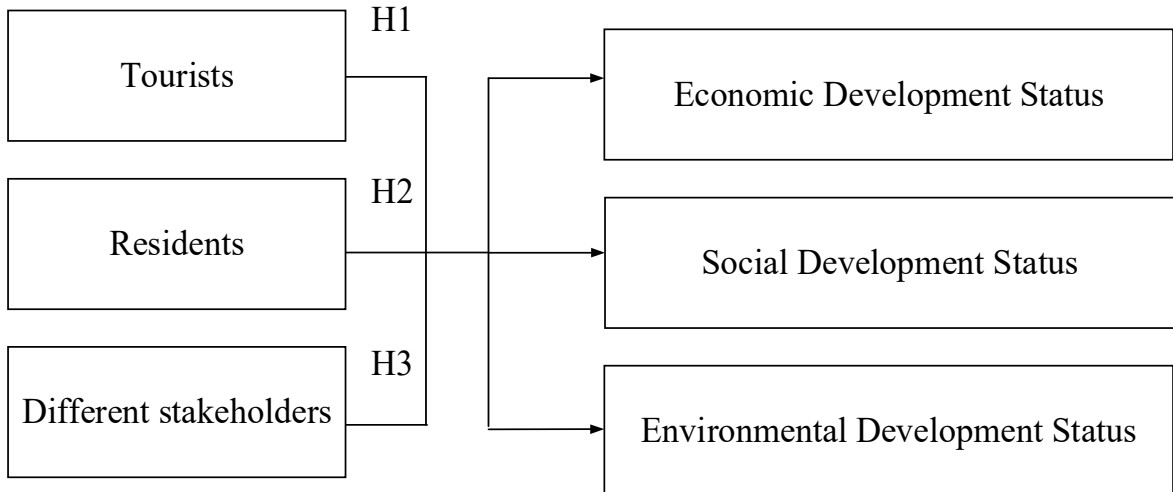

**Figure 2.** Study framework.

According to the research framework, we propose the following hypothesis:

**Hypothesis 1.** *There is consistency in the recognition of current economic development among different stakeholders.*

**Hypothesis 2.** *There is consistency in the recognition of current social development among different stakeholders.*

**Hypothesis 3.** *There is consistency in the recognition of the current status of environmental development among the different stakeholders.*

### 2.2. Study Procedure and Instruments

Tourism development needs to be considered in a multi-faceted way and the impact of tourism development is more complicated nowadays [6,10,12]. More detailed answers can be obtained through comprehensive surveys [10,11,14,24,28]. Therefore, the authors first reviewed the theories and literature on tourism development [3–32], interpreted the research results [38,39], compiled a 60-question questionnaire on the current status of tourism development, and then discussed with experts and scholars to complete the outline of the questionnaire, which was divided into two parts: background information and tourism development. In addition to the background information on issues such as identity, gender, age, and occupation, tourism development included 57 issues: economy (15), society (21) and environment (21), as shown in Table 1.

**Table 1.** Initial questionnaire issue preparation.

| Level | Issue Content | Number of Questions |
|---|---|---|
| Background | Identity, gender, age | 1–3 |
| Economic | Increase employment opportunities<br>Enhance in consumer prices for tourist activity content<br>Enhance in tourism consumption expenditure costs<br>Tourism development to increase tourism facilities and local characteristic industrial products<br>More diversified tourism industry categories<br>Provide interpretive guides during travel or services for the use of amusement facilities<br>Increase the choice of tourist, leisure and life consumption opportunities<br>Get opportunities for promotion or preferential use of local tourist facilities<br>Enhance scale of community tourism development<br>Public facilities and construction and maintenance quality promotion<br>Tourist consumption health and service quality promotion<br>Establish communication channels between the community and the government<br>Establish protection policies for local tourism development, to prevent monopoly by consortia<br>Mechanisms and norms that can participate in the formulation of tourism development policies<br>Develop DIY or product portfolio or creative products to enhance tourists' willingness to spend | 4–18 |
| Society | Enhance tourism visibility<br>Enhance the quality of tourism services and activities<br>Enhance understanding of the content of community sightseeing activities<br>More treasure the community environment of tourist destinations<br>Sufficient introduction to local tourism related indicators or descriptions<br>Provide more choices of local tourist and leisure facilities<br>More young people are employed locally<br>Increased opportunities to participate in local DIY activities<br>Improving the establishment of hardware and software facilities in local communities<br>Tourism environment and space quality enhance<br>Indigenous cultural traditions and historical sites are preserved<br>Good image of local tourism-related industry business or organizations<br>Intensive promotion and development of traditional cultural activities<br>To feel friendly and trust the local residents<br>Nice interaction among tourists, and between tourists and residents<br>Increased number and visibility of traditional culture and industries<br>tourist activity management and safety maintenance quality enhance<br>Sufficient number of fireman and police, security officer<br>Have a sense of security during leisure travel<br>Willing to travel here again | 19–39 |

**Table 1.** *Cont.*

| Level | Issue Content | Number of Questions |
|-------|---------------|---------------------|
| Surroundings | The community's natural environment is maintained cleaner<br>The environmental literacy concept of the villagers is good, do not throw away trash<br>Historic buildings are preserved<br>Willing to attach importance to and participate in the conservation of local natural ecology<br>Willing to cooperate with environmental protection announcement slogans or manuals, care for the environment<br>The tourist trail has a complete appearance and smooth movement<br>The bicycle lane facilities are well preserved and circulation is best<br>Public transportation construction is helpful for personal travel round trip<br>Wireless Wi-Fi online provides tourist and travel information inquiry<br>It is convenient to rent bicycle facilities and location planning<br>To feel smooth when moving in transportation or round trip<br>Increase in the size of community construction area and the number of facilities<br>Sufficient planning of parking and pavilion leisure facilities<br>Tourist will affect the quality of the local natural environment<br>Increase leisure and experience space in tourist tours<br>The local water source is polluted<br>Local air quality is nice<br>Good quality of water for people's livelihood in tourist<br>The development area of hillside vegetation and forest land in the village increased<br>The artificial landscape area increased<br>Emission intensive of fumes from car and motorcycle during tourism | 40–60 |

Except for background information, all issues were measured on a 5-point Likert-type scale, with 1 indicating strongly disagree and 5 indicating strongly agree. The questionnaire was first compiled by referring to the relevant literature, then reviewed by three experts for content validity, and 50 questionnaires were distributed. The results were analyzed using SPSS 22.0 statistical software and then tested with statistical methods. When Kaiser–Meyer–Olkin (KMO) > 0.06 and the $p$-value of the Bartlett test was less than 0.01 ($p < 0.01$), the scale was suitable for continuous factor analysis [40]. A coefficient $\alpha$ greater than 0.60 indicates that the questionnaire has good reliability [41]. The results of the analysis are shown in Table 2.

**Table 2.** Analysis of the Tingxi Reservoir Tourism Development Perception Questionnaire.

| Construct | Subfacet | Issues | Cronbach's α |
|---|---|---|---|
| Economic | People's livelihood price | 1. Increase employment opportunities<br>2. Enhance in land and house prices<br>3. Enhance in expenditure costs | 0.618–0.654 |
| | Industry construction | 1. Combination of local characteristic industries<br>2. Increase tourism industry<br>3. Increase interpretation facilities<br>4. Increase leisure opportunities<br>5. Preferential tourism facilities<br>6. Increase tourism construction | 0.809–0.815 |
| Society | Community development | 1. Maintain complete public facilities<br>2. Enhance medical and health standards<br>3. Establish community communication channels<br>4. Develop protection policy settings<br>5. Participate in tourism policy planning<br>6. Develop creative products | 0.774–0.783 |
| | Village construction | 1. Enhance tourism visibility<br>2. Improve the quality of tourism services<br>3. Participate in community tourism affairs<br>4. Actively clean up the community environment<br>5. Sufficient instructions for local tourism<br>6. Increased selection of leisure facilities | 0.756–0.786 |
| | Atmosphere of life | 1. Youth return to their hometowns for development<br>2. Industry to contribute to local development<br>3. Improve the living environment<br>4. Improve the quality of tourist activities<br>5. Protect the indigenous culture | 0.699–0.728 |
| Environment | Cultural safety | 1. Best image of foreign consortia<br>2. Development of traditional cultural activities<br>3. Tourists feel friendly<br>4. Best interaction among residents<br>5. Invest in the indigenous cultural industry<br>6. Enhance community self-government management<br>7. Sufficient fireman and police, security officer<br>8. Have a sense of security in life<br>9. Willing to travel again or buy a property in the local area | 0.847–0.856 |
| | Conservation measures | 1. Clean community environment<br>2. Do not throw away trash by tourists<br>3. Complete preservation of historical sites<br>4. Participation in nature conservation<br>5. Public environment awareness of environmental literacy | 0.748–0.783 |
| | Leisure environment | 1. Complete tourists trails<br>2. Perfect bicycle lane management<br>3. Public transportation facilitates tourism<br>4. Wi-Fi online coverage<br>5. Cheap bicycle rental<br>6. Complete transportation facilities | 0.781–0.800 |
| | Tourist facility | 1. Increased facility construction area<br>2. Adequate parking and leisure facilities<br>3. Environmental quality affected by tourists<br>4. Adequate personal living space | 0.675–0.715 |

Economy (15) had a KMO > 0.940, a Bartlett approximate $\chi^2$ value of 2386.692, and a degree of freedom (df) of 105 with a significance of 0.000 ($p < 0.001$), making it suitable for factor analysis. Explained variances for the scale were 18.72%, 14.509%, and 14.491% for a total explained variance of 47.72%. All these were retained after factor analysis and taking into account the understanding of the actual state of economic development. The three areas were named: people's livelihood price (3), industrial construction (6), and community development (6). They contained a total of 15 questions and the three scales were 0.729, 0.838, and 0.810, respectively.

Society (21) had a KMO > 0.943, a Bartlett approximate $\chi^2$ value of 3303.559, a degree of freedom (df) of 210, and a significance of 0.000 ($p < 0.001$), thus making it suitable for factor analysis. Explained variances of the scale were 15.187%, 15.094%, and 11.783%, for an overall explained variance

of 42.064%. All these were retained after factor analysis and taking into account the understanding of the actual situation of social development. Three areas were named: village construction (6), atmosphere of life (5), and cultural safety (9). In total, they contained 21 questions with three scales of 0.807, 0.756, and 0.864, respectively.

The environment (21) had a KMO > 0.950, a Bartlett approximate $\chi^2$ value of 3658.093, a degree of freedom (df) of 210, and a significance of 0.000 ($p < 0.001$), making it suitable for factor analysis. Explained variances in the scale were 16.495%, 14.142%, 9.245%, and 7.665%, respectively, for a total explained variance of 47.546%. All of these were retained after factor analysis and taking into account an understanding of the physical conditions of environmental development. The four areas were named: Conservation Measures (5), Leisure Environment (6), Tourism Facilities (4), and Landscape and Ecological Environment (6). They contained a total of 21 questions with three scales of 0.799, 0.817, 0.748, and 0.779, respectively.

Subsequently, the researcher went to the local area to conduct fieldwork in October 2019, but due to the impact of the COVID-19 epidemic, the questionnaire sample collection process and effectiveness were affected. The questionnaire was collected from November 2019 to May 2020. Initially, the questionnaires were collected on site by random sampling, and later, the questionnaires were collected through an online platform in a snowballing fashion, and a total of 804 valid questionnaires were collected. The data were collected and analyzed using the SPSS for Windows 22.0 statistical package in order to statistically check the reliability of the questionnaire issues and analyze the results using descriptive and t-test analysis. Interviews were subsequently conducted to supplement missing information. With the consent of the interviewees, a semi-structured design and open-ended interviews were conducted with five interviewees including tourism practitioners, residents, and academics who had experience in traveling to Tingxi Reservoir or had some knowledge of the current development of the reservoir. Researchers interacted by video, taking the results of the questionnaire analysis as the topic, in order to solicit their opinions on the results of the questionnaire. As shown in Table 3.

**Table 3.** Background information of the interviewees and outline of the interview.

| Identity | Gender | Residence Time/Years of Work Experience | Identity | Gender | Residence Time/Years of Work Experience |
|---|---|---|---|---|---|
| The elderly | Male | 25 | Tourist guide | Male | 40 |
| The elderly | Female | 30 | Tourist guide | Female | 25 |
| Professor | Male | 15 | | | |

| Construct | Issues |
|---|---|
| Impact of tourism development | 1. What impact does tourism development have on the economic, social, and environmental development of the community? 2. According to research and investigation, what causes the impact of economic, social and environmental issues? |

After collecting the opinions of the interviewees, we recorded the interview content and then asked the interviewees to verify the accuracy of the recorded content. The information from the questionnaire was then integrated, and the results were analyzed, and the research paper was constructed with the order of induction, organization, and analysis [16]. The information from the questionnaire was then integrated, and the results were analyzed, and the research paper was constructed with the order of induction, organization, and analysis [16]. Finally, using the multivariate verification and analysis method and combining information from different research subjects, research theories and methods to examine multiple data from multiple viewpoints and to compare the results of various studies [38,40], accurate knowledge and meaning were obtained in order to examine the current situation of promoting tourism development in Tingxi Reservoir.

*2.3. Study Scope and Limitations*

The study aimed to investigate the current tourism development of Tingxi Reservoir by applying a mixed-method research approach using the reservoir as a location, surrounding villages as the range, and local residents and people who had traveled to the reservoir as subjects.

The initial phase of the study began in October 2019, but due to the extensive study area, the research team was unable to complete the sample collection immediately as there were limitations in manpower, resources, and funding during the study period, and the outbreak of COVID-19 in December 2019 further delayed the sample collection process, which took a total of seven months. Although the online questionnaire platform was adopted to collect the information, it was limited by the willingness of the respondents and their proficiency in using 3C products, which led to the shortcomings of the information collected by the researcher. Summing up the above explanations, it is unlikely that more comprehensive information can be obtained due to the limitations of the sample background. If this resulted in any discrepancy in the study, it will be taken into consideration for further study.

## 3. Analysis of Results

A total of 804 samples were obtained. Results showed that most of the sample subjects were residents (67.7%), while only 30.8% were tourists. Most of the samples were from females (75.9%), with males accounting for the least (24.1%). Most of the respondents were aged 21–30 (67.2%), followed by those under 20 years old (20.4%), and the least were aged 51 or above (1.7%).As shown in Table 4.

**Table 4.** Descriptive characteristics of the participants.

| Identity | | | |
|---|---|---|---|
| **Identity** | **Percentage** | **Age** | **Percentage** |
| Residents | 69.7% | Under 20 | 20.4% |
| Tourists | 30.8% | 21–30 | 67.2% |
| **Gender** | **Percentage** | 31–40 | 6.7% |
| Male | 24.1% | 41–50 | 2.2% |
| Female | 75.9% | 51–60 | 1.7% |
| | | Over 61 | 1.7% |

*3.1. Cognitive Analysis of Economic Development*

The questionnaire was revised with reference to previous literature [3–32] where a score of 1 means strongly disagree and 5 means strongly agree. First, the perceptions of residents and tourists were investigated by statistical tests; next, the *t*-test was used to analyze the differences in perceptions among different stakeholders; then the respondents' perceptions were combined, and finally, a multivariate test was used to explore the differences [16,38,40].

The perceptions of the residents and tourists of the current economic development were analyzed, as shown in Table 5. Results showed that residents and visitors agreed only on increasing job opportunities and construction, land, and depressed prices, but disagreed on the rest.

**Table 5.** The perceptions of residents and tourists of the current economic development.

|  | Facets | Highest | M | Lowest | M |
|---|---|---|---|---|---|
| Residents | People's livelihood price | Increase employment opportunities | 3.86 | Land and housing prices are rising Enhanced expenditure costs | 3.69 |
|  | Industry construction | Enhanced tourism construction | 3.87 | Increased leisure opportunities | 3.68 |
|  | Community development | Participate in tourism policy planning | 3.88 | Enhanced medical and health standards | 3.71 |
| Tourists | People's livelihood price | Increase employment opportunities | 4.04 | Land and housing prices are rising Enhanced expenditure costs | 3.74 |
|  | Industry construction | Enhanced tourism construction | 3.96 | Enhanced interpretation facilities | 3.77 |
|  | Community development | Developing creative products | 4.03 | Establish community communication channels | 3.78 |

The inference is that since the government has been actively improving the public infrastructure and utilizing the natural resources of the area to attract investment, it is hoped that this will bring business opportunities and create jobs. Although many villages have been preserved and are suitable for rural and eco-tourism activities, the hinterland of Tingxi Reservoir is large, the industries in the surrounding villages are highly similar, and the population is aging, resulting in poor overall development planning. As a result, residents believe that although there are more opportunities for employment, tourism construction, and participation in tourism development policies, the leisure options are low, the medical and health conditions are poor, and the consumption conditions have not improved.

The diverse nature of the local ecology has created abundant tourism resources and brought different travel experiences. With the existing resources and culture, the development of handicraft products and agricultural products could attract tourists to spend money. However, although the hinterland can be developed and the infrastructure upgraded, there is a shortage of manpower in the industry due to the obvious trend of aging. As a result, tourists believe that development can help increase job opportunities, develop creative products, increase leisure options, and maintain a low consumption level. Nevertheless, due to the sparse population, there is a lack of interpretive facilities, and communication channels in the scenic spots are poor.

In summary, residents and visitors agree that the government's interest in tourism development has resulted in improved local infrastructure, stimulated industrial operations, business opportunities for the villages, and increased employment opportunities, but perhaps imperfect overall development decisions have also resulted in low land prices and low consumer willingness in most areas. In addition, the aging population, shortage of manpower, lack of tourism facilities, and high similarity of industries have led to a divergence of views.

Further exploring the differences in the perceptions of the impact of economic development among different stakeholders, we found that there were significant differences ($p < 0.001$) in the issues of leisure opportunities and health care standards, indicating that different stakeholders had different views on the increase in leisure opportunities and the current development of health care standards, as shown in Table 6.

**Table 6.** An analysis of the differences in the cognition of the current economic development among different stakeholders.

| Issue | | Residents | | Tourists | | T | *p*-Value |
|---|---|---|---|---|---|---|---|
| | | M | SD | M | SD | | |
| People's livelihood price | Increase in employment opportunities | 3.86 | 0.841 | 4.04 | 0.724 | 4.792 | 0.029 |
| | Land and housing prices are rising | 3.69 | 0.939 | 3.74 | 0.809 | 7.901 | 0.005 |
| | Enhanced expenditure costs | 3.69 | 0.928 | 3.78 | 0.763 | 9.035 | 0.003 |
| Industry construction | Local characteristic industry combination | 3.71 | 0.946 | 3.85 | 0.797 | 9.692 | 0.002 |
| | Enhanced tourism construction | 3.75 | 0.854 | 3.91 | 0.788 | 6.143 | 0.014 |
| | Enhanced interpretation facilities | 3.72 | 0.774 | 3.77 | 0.840 | 0.079 | 0.779 |
| | Increased leisure opportunities | 3.68 | 0.963 | 3.96 | 0.747 | 23.521 | 0.000 * |
| | Tourist facility discounts | 3.75 | 0.817 | 3.82 | 0.763 | 4.026 | 0.045 |
| | Enhanced tourism construction | 3.87 | 0.812 | 3.90 | 0.730 | 5.721 | 0.017 |
| Community development | Complete maintenance of public facilities | 3.78 | 0.794 | 3.91 | 0.742 | 3.777 | 0.053 |
| | Enhanced medical and health standards | 3.71 | 0.912 | 3.89 | 0.797 | 10.742 | 0.001 * |
| | Establish community communication channels | 3.74 | 0.781 | 3.78 | 0.760 | 1.082 | 0.299 |
| | Development and protection policy settings | 3.75 | 0.830 | 3.86 | 0.773 | 3.131 | 0.078 |
| | Participate in tourism policy planning | 3.88 | 0.820 | 3.92 | 0.764 | 4.351 | 0.038 |
| | Developing creative products | 3.86 | 0.878 | 4.03 | 0.784 | 3.562 | 0.060 |

* = $p < 0.001$.

It is inferred that due to the ecological diversity of Tingxi Reservoir, the village is rich in tourism resources and agricultural specialties, but the similarity between existing tourism resources and industries is high, making it less attractive to tourists and unable to meet the changing tourism needs. In addition, due to the inconvenient transportation and the aging population, the local medical resources are only sufficient to meet the needs of tourists, and not residents. As a result, different stakeholders have different views on leisure opportunities and the current development of medical and health care facilities. As shown in Figure 3.

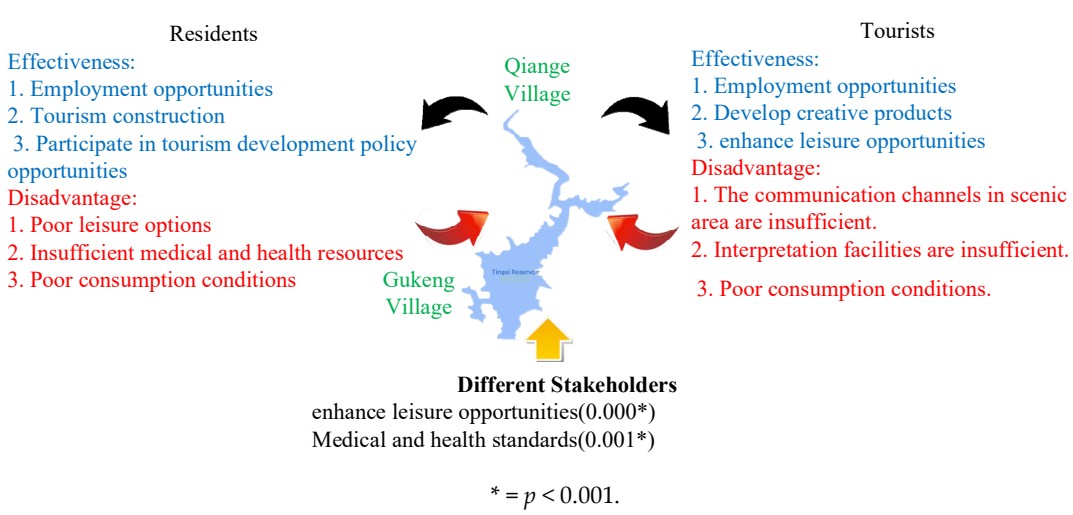

* = $p < 0.001$.

**Figure 3.** Perception analysis of the current economic development.

*3.2. Cognitive Analysis of Social Development*

The perceptions of residents and tourists on the impact of social development were first investigated separately, as shown in Table 7. The results showed that residents and tourists shared the same views on the residents' initiative to clean up the community environment, the youths' low willingness to return to their hometowns, and the shortage of police and fire safety personnel, but differed in all other aspects.

**Table 7.** The perceptions of the residents and tourists of the current social development.

| | Facets | Highest | M | Lowest | M |
|---|---|---|---|---|---|
| Residents | Village construction | Participate in community tourism affairs | 3.94 | Increased choice of leisure facilities Actively organize the community environment | 3.80 |
| | Atmosphere of life | Improve the living environment | 3.89 | Youth back to the country development | 3.73 |
| | Culture safety | Feel safe in life | 3.87 | Sufficient fireman and police, security officer | 3.68 |
| Tourist | Village construction | Increased choice of leisure facilities | 3.94 | Actively sort out community environment | 3.84 |
| | Atmosphere of life | Industry gives back to local development | 3.94 | Youth back to the country | 3.75 |
| | Culture safety | Development of traditional cultural activities | 3.93 | Sufficient fireman and police, security officer | 3.63 |

It is inferred that, since improving local conditions, enhancing safety and well-being, and meeting people's expectations are the development goals of the policy, combining the existing resources and residents' manpower to jointly promote tourism development, attract tourists to spend money, and promote the village economy is the expectation for the development of remote areas. However, the vast expanse of rural areas is not easy to develop and manage, resulting in a declining population, a lack of mobility for the elderly, and a shortage of human resources, creating a manpower gap in the industry, which affects investment intentions and reduces job opportunities. As a result, residents believe that the development will help improve their living environment, give them a sense of security, and increase their opportunities to participate in community tourism activities. However, residents are not willing to participate in community development, there are few leisure facilities to choose from, there is a lack of police and fire safety manpower, and young people are not willing to return to their hometowns for career development.

The natural ecological richness of the water source area makes the village very attractive to tourists. The government is willing to build tourism facilities and improve the local community environment to encourage investment from enterprises. However, in the early stages of development, job opportunities were scarce, forcing young people to look for jobs elsewhere. In addition, villagers rely on traditional industries to make a living, and the high workload, coupled with the aging of the population, makes it impossible to meet the demand for manpower for industrial development and community maintenance in villages or scenic areas. As a result, tourists believe that after development, the industry can give back to the local community and promote traditional cultural activities to increase the choice of recreational facilities, but the residents have a low awareness of community development, resulting in a low willingness of young people to return to their hometowns for development and a shortage of police and fire safety officers.

Further exploring the differences in perceptions of the impact of social development among different stakeholders revealed that there were significant differences ($p < 0.001$) on the issue of traditional cultural activities, indicating that different stakeholders had different views on the effectiveness of the development of traditional cultural activities, as shown in Table 8.

**Table 8.** Analysis of the differences in perceptions of social development status among different stakeholders.

| Issue | | Residents | | Tourists | | T | *p*-Value |
|---|---|---|---|---|---|---|---|
| | | M | SD | M | SD | | |
| Village construction | Enhance tourism visibility | 3.92 | 0.862 | 3.91 | 0.823 | 3.113 | 0.078 |
| | Improve the quality of tourism services | 3.85 | 0.858 | 3.86 | 0.739 | 4.493 | 0.035 |
| | Participate in community tourism affairs | 3.94 | 0.827 | 3.87 | 0.743 | 1.903 | 0.169 |
| | Actively sort out community environment | 3.81 | 0.928 | 3.84 | 0.808 | 5.848 | 0.016 |
| | Sufficient tourist pointer | 3.84 | 0.909 | 3.79 | 0.766 | 4.175 | 0.042 |
| | Increased choice of leisure facilities | 3.81 | 0.833 | 3.94 | 0.761 | 4.393 | 0.037 |
| Atmosphere of life | Youth back to the country | 3.73 | 0.995 | 3.75 | 0.926 | 2.068 | 0.151 |
| | Industry gives back to local development | 3.79 | 0.865 | 3.94 | 0.752 | 5.079 | 0.025 |
| | Improve the living environment | 3.89 | 0.828 | 3.93 | 0.740 | 4.708 | 0.031 |
| | Enhance the quality of sightseeing activities | 3.80 | 0.909 | 3.92 | 0.793 | 7.156 | 0.008 |
| | Protect the indigenous culture | 3.88 | 0.845 | 3.91 | 0.811 | 0.654 | 0.419 |
| Culture safety | best image of foreign business | 3.75 | 0.890 | 3.76 | 0.741 | 7.019 | 0.008 |
| | Development of traditional cultural activities | 3.78 | 0.892 | 3.93 | 0.744 | 15.780 | 0.000 * |
| | Tourists feel friendly | 3.79 | 0.828 | 3.79 | 0.779 | 2.352 | 0.126 |
| | best interaction among residents | 3.81 | 0.882 | 3.80 | 0.788 | 5.969 | 0.015 |
| | Invest in unique cultural industries | 3.82 | 0.878 | 3.88 | 0.720 | 9.088 | 0.003 |
| | Community autonomy management rises | 3.74 | 0.869 | 3.83 | 0.740 | 8.944 | 0.003 |
| | Sufficient fireman and police, security officer | 3.68 | 0.919 | 3.63 | 0.821 | 3.271 | 0.071 |
| | Feel safe in life | 3.87 | 0.812 | 3.80 | 0.739 | 2.932 | 0.088 |
| | Willing to travel or buy a local | 3.79 | 0.776 | 3.80 | 0.846 | 0.341 | 0.560 |

\* = *p* < 0.001.

As a result of technological and civilizational advances and the evolution of lifestyles, it is inferred that traditional village culture and customs, which allow visitors to experience the characteristics of ancient civilizations, have become a special tourist image and attraction. However, the culture and customs of the villages have been with the inhabitants for their entire lives, and although they have a unique culture, they still need the construction and knowledge of modern civilization and technology to improve the quality of life of the inhabitants. This has led to different views on the effectiveness of traditional cultural activities among different stakeholders. As shown in Figure 4.

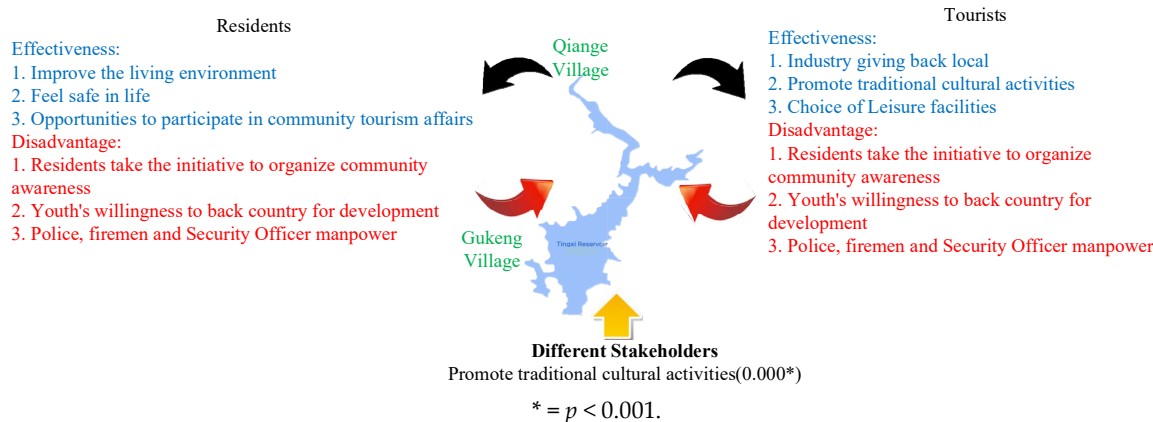

**Figure 4.** Perception analysis of current social development.

### 3.3. Cognitive Analysis of Environmental Development

The views of residents and tourists on the impact of environmental development were analyzed separately, as shown in Table 9, and it was found that residents and tourists only agreed on the wide coverage of a Wi-Fi network, good air quality, and not littering from the tourists, while the rest were different.

**Table 9.** Perception of the current state of environmental development by residents and visitors.

|  | Facets | Highest | M | Lowest | M |
|---|---|---|---|---|---|
| Residents | Conservation measures | Historic monuments are kept intact | 3.95 | Tourists do not throw away garbage | 3.78 |
|  | leisure environment | Wide Wi-Fi online coverage | 3.87 | Bicycle rental is cheap | 3.63 |
|  | Tourist facility | Environmental quality is affected by tourists | 3.95 | Sufficient personal living space | 3.69 |
|  | Landscape and ecological environment | Best air quality | 3.93 | Pollution of water quality | 3.63 |
| Tourist | Conservation measures | Participate in nature conservation | 3.97 | Tourists do not throw away garbage | 3.73 |
|  | leisure environment | Public transport helps travel Wide Wi-Fi online coverage | 3.93 | Perfect bicycle lane management | 3.79 |
|  | Tourist facility | Sufficient personal living space | 3.91 | Adequate parking and leisure facilities | 3.81 |
|  | Landscape and ecological environment | best air quality | 3.90 | Car and motorcycle oil fume pollution | 3.67 |

It is inferred that the government's commitment to construction, improving the efficiency of the nation's Internet, advancing technology, civilization and quality of life as well as preserving the natural ecology and environment and protecting village culture and historical buildings has helped this area become a major tourist attraction. However, the influx of tourists attracts huge business opportunities, and local business owners try to inflate prices to seek huge profits; in addition, the hinterland is vast, construction funds are limited, and the existing parking facilities cannot meet the demand; moreover, the spending power of tourists has increased dramatically, but the quality is difficult to control, and the amount of garbage is large, which affects the environment. Therefore, residents believe that after development, historical monuments are preserved intact, Wi-Fi network coverage is wide, and the water and air quality is good, but the amount of garbage and environmental quality is affected by tourists, bicycle rental is expensive, and their living space is insufficient.

Since air quality, clear water sources, and a beautiful natural environment are the main appeal of Tingxi Reservoir, the government is willing to invest in upgrading Internet services and improving transportation. The residents are willing to work together to protect the natural ecology in order to attract tourists and improve the economic situation. However, crowds are booming, the amount of tourist waste is increasing, and the environmental literacy of tourists varies. In addition, due to the vast land area and the lack of funds for village construction, the existing bicycle facilities and parking space planning cannot meet the needs of a large number of tourists. As a result, tourists think that after development, there is a high level of participation in nature conservation, that public transportation facilitates tourism, the Wi-Fi network coverage is wide, personal living space is sufficient, there are few vehicles, air quality is good, but that tourists litter, bicycle lanes are not well managed, and parking rest facilities are insufficient.

The differences in perceptions of the impact of economic development among different stakeholders were found to be significant ($p < 0.001$), indicating that different stakeholders had different perceptions of the current status of living space planning and development, as shown in Table 10.

**Table 10.** Analysis of the differences in awareness of the current state of environmental development among different stakeholders.

| | Issue | Residents | | Tourists | | T | *p*-Value |
|---|---|---|---|---|---|---|---|
| | | M | SD | M | SD | | |
| Conservation measures | To clean community environment | 3.91 | 0.792 | 3.88 | 0.769 | 0.163 | 0.687 |
| | Tourists do not throw away garbage | 3.78 | 0.869 | 3.73 | 0.757 | 3.736 | 0.054 |
| | Historic monuments are kept intact | 3.95 | 0.834 | 3.94 | 0.776 | 4.227 | 0.040 |
| | Participate in nature conservation | 3.92 | 0.786 | 3.97 | 0.760 | 1.512 | 0.220 |
| | Public cognition of environmental literacy | 3.87 | 0.772 | 3.92 | 0.746 | 0.387 | 0.534 |
| Leisure environment | Complete tourist trail | 3.86 | 0.854 | 3.84 | 0.762 | 1.814 | 0.179 |
| | Perfect bicycle lane management | 3.74 | 0.950 | 3.79 | 0.788 | 4.282 | 0.039 |
| | Public transport helps travel | 3.80 | 0.931 | 3.93 | 0.769 | 6.832 | 0.009 |
| | Wide Wi-Fi online coverage | 3.87 | 0.772 | 3.93 | 0.800 | 0.186 | 0.666 |
| | Bicycle rental is cheap | 3.63 | 0.882 | 3.88 | 0.800 | 4.883 | 0.028 |
| | Perfect transportation line facilities | 3.86 | 0.816 | 3.85 | 0.757 | 0.888 | 0.346 |
| Tourist facility | Increased facility construction area | 3.88 | 0.767 | 3.87 | 0.727 | 0.462 | 0.497 |
| | Adequate parking and leisure facilities | 3.74 | 0.869 | 3.81 | 0.834 | 2.084 | 0.150 |
| | Environmental quality is affected by tourists | 3.95 | 0.846 | 3.88 | 0.785 | 1.377 | 0.241 |
| | Sufficient personal living space | 3.69 | 0.834 | 3.91 | 0.730 | 10.397 | 0.001 * |
| Landscape and ecological environment | Pollution of water quality | 3.63 | 0.893 | 3.77 | 0.766 | 7.078 | 0.008 |
| | best air quality | 3.93 | 0.807 | 3.90 | 0.758 | 0.592 | 0.442 |
| | Vegetation forest land has been developed | 3.81 | 0.858 | 3.81 | 0.767 | 3.656 | 0.057 |
| | Destroy the original habitat | 3.77 | 0.823 | 3.87 | 0.773 | 3.531 | 0.061 |
| | Car and motorcycle oil fume pollution | 3.82 | 0.817 | 3.67 | 0.838 | 0.167 | 0.683 |

\* = $p < 0.001$.

It can be inferred that while residents expected the development to bring economic growth, increase their income, and improve their quality of life, the influx of tourists has taken over their living space and affected their daily routines. Although there are few scenic spots in the villages and not enough tourism infrastructure and industries to accommodate the huge number of tourists, visitors still look forward to a comfortable environment and space away from the hustle and bustle of the city, where they can relax both physically and mentally. Therefore, although the land is vast and the environment is spacious, the definition of living space varies according to the conditions and needs of individuals in different roles and positions. As a result, different stakeholders had different views on the current situation of adequate personal living space. As shown in Figure 5.

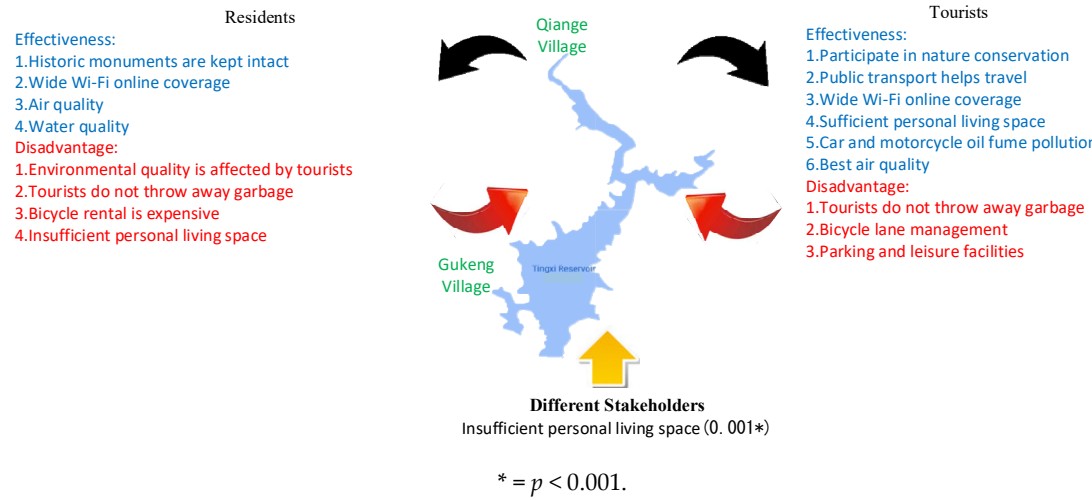

\* = $p < 0.001$.

**Figure 5.** Perception analysis of current environmental developments.

## 4. Conclusions and Recommendations

### 4.1. Conclusions

Research believes that the current tourism development can help villages and residents around the reservoir to obtain employment opportunities and construction, increase the residents' willingness to participate in policies, improve the quality of life and the environment, protect cultural and historical relics, improve network information systems, and will not harm the air quality. However, it does not help increase the land and housing prices, develop leisure activities, and improve conditions for medical facilities, coupled with increased consumption costs, environmental damage caused by tourists, and insufficient police and firefighting personnel, which does not help attract young people to return to their homes.

Although development has enabled the region to obtain company resources, increase employment opportunities, develop creative products, increase leisure options, promote traditional cultural activities, increase people's awareness of ecological protection, improve transportation and network facilities, and maintain low consumption level, spacious area, good air quality, and ample living space, which ttract tourists to travel and consume, however, the shortage of interpreters, parking and entertainment facilities, poor communication channels, poor management of bicycle facilities and waste management as well as the low awareness of participation of residents have led to insufficient manpower and low consumption willingness of young tourists.

At present, different stakeholders have different opinions on the development effectiveness of leisure opportunities, medical care, traditional cultural activities and residential space planning.

### 4.2. Recommendations

#### 4.2.1. The resident population

It is necessary to emphasize tourism development, invest in tourism industry planning, stabilize commodity prices, and bring in manpower to satisfy the tourism demand, in order to alleviate hardship and attract young people to return home.

#### 4.2.2. The tourists

It is necessary to improve personal environmental literacy, carry out waste separation, reduce the production of tourist waste, and work together to protect the environment in order to preserve the beautiful scenic environment.

#### 4.2.3. The government

Invest funds in public facilities, provide channels for tourists to lodge complaints, develop scenic spots to divert tourists, introduce enterprises and technologies to develop new industries, and train interpretive talents to provide in-depth tourism services.

#### 4.2.4. Research suggestions

It is suggested that the follow-up study should understand the travel behavior of tourists, analyze the impact of development on the current leisure behavior of residents, investigate the tourism resource potential of the reservoir, and finally discuss the impact of development on neighboring villages to complete the relevant research information.

**Author Contributions:** C.-C.S. research information distribution and resource input. H.-H.L. conceived, designed and wrote this paper. C.-H.H. and J.-H.C. help check spelling. J.-H.C. helped check the grammar of the article. C.-F.L. data curation, writing-review & editing. All authors have read and agreed to the published version of the manuscript.

**Funding:** This research received no external funding.

**Conflicts of Interest:** The authors declare no conflict of interest.

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
