# Peer review of "Research on the Impact of Tourism Development on the Sustainable Development of Reservoir Headwater Area Using China’s Tingxi Reservoir as an Example"

_water, doi:10.3390/w12123311_

Round 1

Reviewer 1 Report

The paper is interesting and the topic is worthy of investigation.

Please add scientific contribution of the paper in abstract and in conclusion part.

Some parts of the paper are unclear:

  • line 23 explain what is Triangulation?
  • line 43 Figure 1 ?
  • line 45 the effectiveness of tourism development ?
  • line 133 …are water quality ecology, water quantity, power generation…?
  • line 152 verification “40”?
  • line 153 construct “16” ?
  • line 231 instead of defects put limitations

The discussion lacks of some critical reflection about the topic, please explain which impact is having the current situation according to the results? Which solution should be considered?

Why you use 60 questions in the questionnaire?

Improve the visibility of Figure 3 and figure 4.

Authors should carefully read the journal standards, the order of references should be checked.

Author Response

Reviewer 1
Comments and Suggestions for Authors
The paper is interesting and the topic is worthy of investigation.
Please add scientific contribution of the paper in abstract and in conclusion part.
Some parts of the paper are unclear:
• line 23 explain what is Triangulation?
Dear reviewer:
We have revised to "multivariate verification method"
• line 43 Figure 1 ?
Dear reviewer:We have revised to "as shown in Figure 1"
• line 45 the effectiveness of tourism development ?
Dear reviewer:
We have revised to "which shows the effectiveness of tourism development"
• line 133 …are water quality ecology, water quantity, power generation…?
Dear reviewer:
We have revised to "Most of the studies that focus on Tingxi Reservoir are about water quality and ecology, water flow, water quality, and power generation"
• line 152 verification “40”?
Dear reviewer:
We have revised to "comparing and verifying data [40]". it is a references.
• line 153 construct “16” ?
Dear reviewer:
We have revised to " [16]". it is a references.
• line 231 instead of defects put limitations
Dear reviewer:
We have polished up the wording of this paragraph.
The discussion lacks of some critical reflection about the topic, please explain which impact is having the current situation according to the results? Which solution should be considered?
Dear reviewer:
We have greatly revised the summary and conclusion.
Why you use 60 questions in the questionnaire?
Dear reviewer:
We have greatly revised the summary and conclusion. And present on line 158
Improve the visibility of Figure 3 and figure 4.
Dear reviewer:
We have adjusted the table.
Authors should carefully read the journal standards, the order of references should be checked.
Dear reviewer:
We have corrected the literature drastically.

Thank you for your suggestions, your suggestions make the article more complete.
I wish you success and look forward to a good reply

Reviewer 2 Report

The authors present a potentially interesting study on the impact of tourist development. Yet the paper is difficult to read and assess due to poor use of the English language. Some specific comments. The method is not described in a trackable way, specific examples: -how was the semi structured interview organised? -how did the input of experts actually work -it does not become clear how the cognitive analysis works. -... Results (specifically Table 5) should be explained more in depth in the text: what are the conclusions that the reader should draw from this table? I would be happy to properly assess a revised version of this manuscript.

Author Response

Reviewer 2

The authors present a potentially interesting study on the impact of tourist development. Yet the paper is difficult to read and assess due to poor use of the English language.

Dear reviewer:

We have greatly revised and corrected the full text in English.

Some specific comments. The method is not described in a trackable way, specific examples: -how was the semi structured interview organised?

Dear reviewer:

We  will interact by video, taking the results of the questionnaire analysis as the topic, in order to solicit their opinions on the results of the questionnaire.

-how did the input of experts actually work -it does not become clear how the cognitive analysis works.

Dear reviewer:

After collecting the opinions of the interviewees, we recorded the interview content and then asked the interviewees to verify the accuracy of the recorded content. The information from the questionnaire was then integrated, and the results were analyzed, and the research paper was constructed with the order of induction, organization, and analysis.

-... Results (specifically Table 5) should be explained more in depth in the text:what are the conclusions that the reader should draw from this table?

Dear reviewer:

We have already conducted a large-scale discussion in Table 5 (now Table 6).

I would be happy to properly assess a revised version of this manuscript.

Thank you for your suggestions, your suggestions make the article more complete.

I wish you success and look forward to a good reply

Round 2

Reviewer 1 Report

Accept in present form.

Author Response

Dear reviewer
Thanks for your suggestions, so that the article can be improved.
wish you all the best

Reviewer 2 Report

I think the readability of the manuscript improved significantly compared to the last version. Yet I find it still difficult to track how the questionaire was developed based on the literature. I would suggest more reverse outlining in which the key literature that is used for the questionaire is summarised in a more systematic way in the introduction.

My second major point it that in the results you are vague about interference because often the phrase it is interfered is used.

Third I think the paper would benefit from a more systematic discussion on the weaknesses of the methodology and the potential impact on the conclusions.

I wish you good luck with finalizing this interesting research.

Author Response

Review 2 – r2

I would suggest more reverse outlining in which the key literature that is used for the questionaire is summarised in a more systematic way in the introduction.

Dear reviewer

Thanks for your suggestion, we have made adjustments. Such as lines 127-149.

My second major point it that in the results you are vague about interference because often the phrase it is interfered is used.

Dear reviewer

Thanks for your suggestion, we have made adjustments. Such as lines 305-310.

Third I think the paper would benefit from a more systematic discussion on the weaknesses of the methodology and the potential impact on the conclusions.

Dear reviewer

Thank you for your suggestion, we have made a supplementary explanation. Such as lines 165-181 and lines 432-448.

All in all, I am very grateful to the reviewers for their suggestions to improve the manuscript.
wish you all the best,
